# Oscillatory phase separation in giant lipid vesicles induced by transmembrane osmotic differentials

**Kamila Oglęcka[1,2], Padmini Rangamani[3†], Bo Liedberg[2], Rachel S Kraut[1], Atul N Parikh[2,4,5]\***

[1]Division of Molecular Genetics and Cell Biology, School of Biological Sciences, Nanyang Technological University, Nanyang, Singapore; [2]School of Materials Science and Engineering, Nanyang Technological University, Nanyang, Singapore; [3]Department of Molecular and Cellular Biology, University of California, Berkeley, Berkeley, United States; [4]Department of Biomedical Engineering, University of California, Davis, Davis, United States; [5]Department of Chemical Engineering and Materials Science, University of California, Davis, Davis, United States

**\*For correspondence:** anparikh@ucdavis.edu

**Present address:** [†]Department of Mechanical and Aerospace Engineering, University of California, San Diego, La Jolla, United States

**Competing interests:** The authors declare that no competing interests exist.

**Reviewing editor**: Randy Schekman, Howard Hughes Medical Institute, University of California, Berkeley, United States

**Abstract** Giant lipid vesicles are closed compartments consisting of semi-permeable shells, which isolate femto- to pico-liter quantities of aqueous core from the bulk. Although water permeates readily across vesicular walls, passive permeation of solutes is hindered. In this study, we show that, when subject to a hypotonic bath, giant vesicles consisting of phase separating lipid mixtures undergo osmotic relaxation exhibiting damped oscillations in phase behavior, which is synchronized with swell–burst lytic cycles: in the swelled state, osmotic pressure and elevated membrane tension due to the influx of water promote domain formation. During bursting, solute leakage through transient pores relaxes the pressure and tension, replacing the domain texture by a uniform one. This isothermal phase transition—resulting from a well-coordinated sequence of mechanochemical events—suggests a complex emergent behavior allowing synthetic vesicles produced from simple components, namely, water, osmolytes, and lipids to sense and regulate their micro-environment.

## Introduction

Giant unilamellar vesicles (GUVs) are the simplest cell-like closed compartments consisting of semi-permeable flexible shells (4–6 nm thick, 5–50 µm diameter), isolating femto- to pico-liter quantities of aqueous core from the surrounding bulk (*Walde et al., 2010*). Although water permeates readily across the vesicular walls ($10^{-2}–10^{-3}$ cm/s) (*Fettiplace and Haydon, 1980*), passive permeation of solutes is significantly lower across the intact membrane (*Deamer and Bramhall, 1986*). As a result, osmotic differentials are readily established between the compartmentalized volume and the surrounding free bath. This in turn triggers a relaxation process, which acts to reduce the osmotic pressure difference across the closed semi-permeable membrane by influx (or efflux) of water depending on the sign of the pressure differential. Thus, for osmolyte-loaded vesicles in a hypotonic environment, water permeates and vesicle swells, until the internal Laplace pressure compensates the osmotic pressure, increasing its volume to surface area ratio (*Ertel et al., 1993*). In this same vein, efflux of compartmentalized water from vesicles embedded in hypertonic media, conversely, decreases the volume to surface area ratio (*Boroske et al., 1981*). Furthermore, because of their large area expansion moduli ($10^2–10^3$ mN m$^{-1}$) and low bending rigidities ($10^{-19}$ Nm), vesicular shells bend readily but tolerate only a limited area of expansion (~5%) (*Needham and Nunn, 1990*; *Hallett et al., 1993*; *Seifert, 1997*). Consequently, GUVs experiencing solute concentration difference across their vesicular boundary

**eLife digest** All living cells are surrounded by a membrane that water can pass through. However, water often contains other molecules called solutes, and many of these cannot pass through the cell membrane. If the concentration of solutes outside the cell is, say, suddenly decreased, then water molecules will tend to move into the cell to lower the solute concentration there. This process, which is called osmosis, strives to equalize the solute concentrations inside and outside the cell.

Osmosis can have dramatic consequences for cells. Animal cells need to be bathed in water to survive, but if the solute concentration outside a cell is higher than inside, the cell can lose a lot of water and die. And if the solute concentration outside is lower, then water enters the cell and it can burst. Single celled microbes use a variety of strategies to counter the movement of water by osmosis: strong cell walls prevent the cell from swelling too much, and channel proteins in the membrane can be opened to allow solutes to pass through. But it is not known how more primitive cells—cells that lived billions of years ago—might have responded to fluctuations in their environment.

Oglęcka et al. have now used artificial membranes to make closed compartments called giant vesicles that mimic certain properties of cells. When giant vesicles are filled with a sugar solution and placed in water with a lower concentration of sugar, a series of events takes place that can lead to the sugar concentration inside and outside the vesicle becoming more equal.

At first the vesicle expands as water enters. However, as the membrane stretches, a temporary hole opens up, which allows some of the excess solute molecules and water to escape, shrinking the vesicle. This sets up cycles of vesicle expansion and contraction that gradually lead to the solute concentrations on both sides of the membrane becoming more equal. This cyclical expansion and contraction of the vesicle also changes the membrane, decorating it with "domains" of specialized molecules, when expanded and uniform, when shrunk.

It is possible that this process may have helped the first primitive cells to survive and, maybe, even benefit from changes in solute concentration in their environment.

adjust their volume, deforming in hypertonic media and swelling in hypotonic ones (*Boroske et al., 1981*; *Ertel et al., 1993*).

A consequence of the osmotic influx of water in vesicles embedded in hypotonic media is the build-up of lateral membrane tension due to changes in the balance of forces within the bilayer producing high energy states (compared to isotonic relaxed vesicles) (*Needham and Nunn, 1990*). Beyond a threshold tension, rupture and pore formation become energetically favorable, lysing the GUVs at lateral tensions corresponding to ~30–40 mNm$^{-1}$ (*Needham and Nunn, 1990*; *Ertel et al., 1993*; *Mui et al., 1993*). The lytic process, however, is not catastrophic; rather it follows a step-wise sequence of events (*Ertel et al., 1993*; *Peterlin et al., 2012*). During each membrane rupture event, only a fraction of the intravesicular solute (and water) is released before the bilayer reseals leaving the vesicle hyper-osmotic with a lower osmotic differential. This then prompts subsequent events of water influx, vesicle swelling, and rupture until sufficient intravesicular solute has been lost, and the membrane is able to withstand the residual sub-lytic osmotic pressure without collapsing (*Wood, 1999*). Thus, GUVs in hypotonic media exhibit oscillations in their sizes—characterized by alternating modulations of vesicular volume, tension, and solute efflux—prompted by repeated cycles of swelling and bursting (*Sandre et al., 1999*; *Karatekin et al., 2003b*; *Popescu and Popescu, 2008*; *Peterlin et al., 2012*).

In the work reported here, we show that the swell–burst cycles in hypertonic vesicles consisting of domain-forming lipid mixtures (*Baumgart et al., 2003*; *Veatch and Keller, 2005*) become coupled with the membrane's compositional degrees of freedom, producing a long-lived transient response characterized by damped oscillations in phase behavior at the membrane surface, cycling between the state characterized by large microscopic domains at the membrane surface and an optically uniform one. This oscillatory phase separation occurs isothermally, and it is driven by a sequence of elementary biophysical processes involving cyclical changes in osmotic pressure, membrane tension, and poration, which attend swell–burst cycles (*Koslov and Markin, 1984*; *Mui et al., 1993*; *Popescu and Popescu, 2008*): in the swelled state, osmotic pressure and elevated membrane tension due to the influx of water promote the appearance of microscopic domains (*Akimov et al., 2007*; *Ayuyan and Cohen, 2008*; *Hamada et al., 2011*). During the burst phase, solute leakage through short-lived membrane

poration (*Sandre et al., 1999*; *Brochard-Wyart et al., 2000*; *Karatekin et al., 2003b*) relaxes the osmotic pressure and membrane tension, breaking up the domains producing an optically uniform membrane. This cyclical pattern does not persist indefinitely: a step-wise diminution of the osmotic pressure differential, because of the solute leakage during burst events, gradually dampens the oscillations ultimately equilibrating the GUV to the residual osmotic differential.

## Results and discussion

### Phase separation in osmotically stressed vesicles

The GUVs (*Morales-Penningston et al., 2010*) we investigated consist of ternary lipid mixtures composed of cholesterol (Ch), sphingomyelin (SM), and the unsaturated phospholipid, POPC (1-palmitoyl-2-oleoyl-*sn*-1-glycero-3-phosphocholine) at room temperature (25°C) ('Materials and methods'). Depending on precise composition and temperature, these mixtures are known to form a uniform single phase or exhibit microscopic phase separation (*Veatch and Keller, 2005*), including one characterized by two co-existing liquid phases: a dense phase enriched in SM and Ch designated as the $L_o$ (liquid-ordered) phase and a second, less dense $L_d$ (liquid-disordered) phase consisting primarily of POPC. To discriminate between the $L_o$ and $L_d$ phases by fluorescence microscopy, we doped our GUVs with a small concentration (0.5 mol%) of a phase sensitive probe, N-lissamine rhodamine palmitoylphosphatidyl-ethanolamine (Rho-DPPE) (*Baumgart et al., 2007*). For an equimolar lipid (1:1:1) composition, the phase diagram predicts the absence of large microscopic domain formation at optical length scales (*Veatch and Keller, 2005*). Consistent with this prediction, our GUVs encapsulating 200 mM sucrose appear optically homogeneous at room temperature in an osmotically balanced, isotonic medium also containing 200 mM sucrose (*Figure 1A,C*). Moreover, they exhibit a flaccid, undulating surface topography (*Video 1*) confirming bending-dominated shape fluctuations (*Seifert, 1997*).

Diluting the extra-vesicular dispersion medium with deionized water produces a hypotonic bath depleted in osmolytes, subjecting the GUVs to a trans-bilayer osmotic differential. A representative fluorescence image obtained shortly after imposing the concentration difference (~200 mM, t < 60 s, reveals that the flaccid topography [*Figure 1A,C*, *Figure 1—figure supplement 1*] and uniform dye distribution of isotonic GUVs are abandoned, replaced by a swollen, spherical boundary and a heterogeneous fluorescence pattern characterized by microscopic, probe-enriched domains consistent with earlier reports) (*Figure 1B,D*) (*Baumgart et al., 2003*; *Hamada et al., 2011*; *Oglecka et al., 2012*). Because Rho-DPPE partitions preferentially into the $L_d$ phase, the appearance of bright domains indicates microscopic $L_d$ phase fluid domains in the $L_o$ phase surroundings.

### Dynamics of phase separation in osmotically swollen vesicles

A time-lapse video of a vesicular population subject to hypotonic conditions (*Figure 2A*, *Video 2*) reveals that the phase separation is not static: optically homogeneous vesicles observed at a given instance break up into surface patterns consisting of large microscopic domains and conversely, those textured by domains adopt an optically homogeneous state over time, with each vesicle undergoing complete single cycles in roughly tens of seconds. Moreover, at any given instance, some vesicles appear homogeneous whereas others are microscopically phase-separated (*Figure 2B* and *Video 3*) producing a heterogeneous landscape. The time-dependent process of the appearance and disappearance of large microscopic domains repeats itself multiple times (n > 10) over several tens of minutes—ultimately producing a steady-state characterized by a fixed microstructure and a rounded boundary. The oscillatory phase separation behavior is fully reproducible for a variety of (1) neutral osmolytes (e.g., glycerol, glucose, lactose, galactose, dextran, sorbitol, and sucrose); (2) GUV sizes (~5–50 μm); (3) initially imposed strengths of osmotic gradients (20–2000 mM); and (4) lipid compositions within the phase co-existence window (*Veatch and Keller, 2005*).

It is known that continuously illuminating single vesicles deliberately using intense light can oxidize lipids (*Ayuyan and Cohen, 2006*) or generate membrane tension by folding the excess membrane area within the localized regions of the enhanced electric field of the light thus suppressing undulations (*Barziv et al., 1995*; *Sandre et al., 1999*). To confirm that the unusual domain dynamics we witness do not result from these effects of optical illumination, we carried out additional experiments. In companion experiments where only occasional low-dose illumination (as opposed to the rapid sequence of illumination used to capture detailed dynamics) at arbitrary time intervals

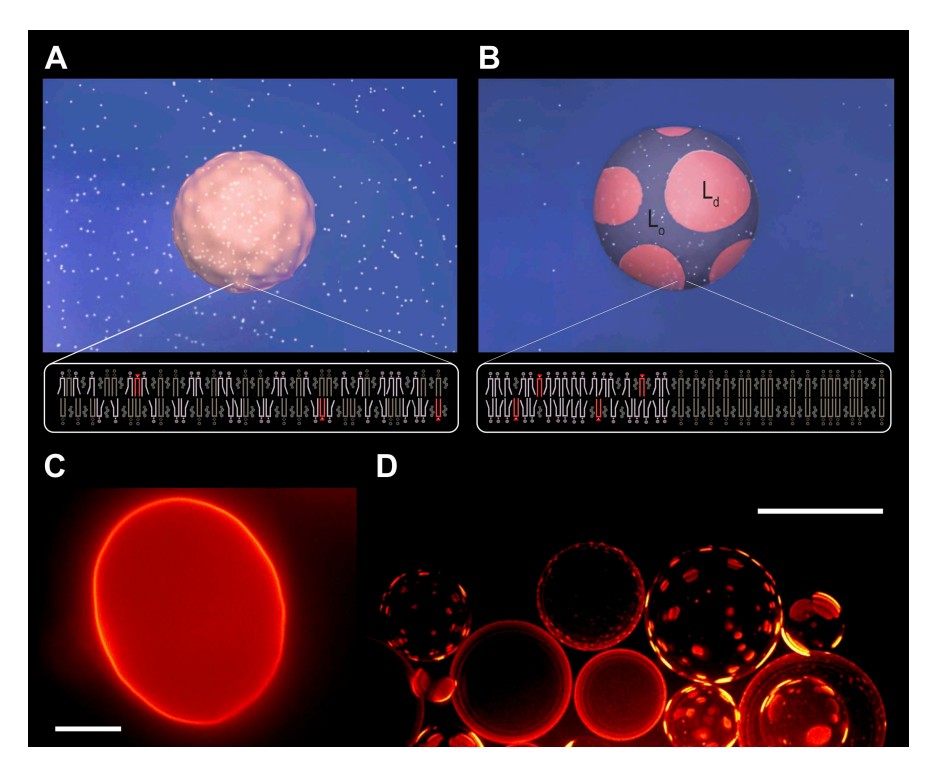

**Figure 1**. Subjecting giant unilamellar lipid vesicles to an osmotic differential. (**A**–**B**), Schematic of a GUV immersed in an osmotically balanced isotonic bath (**A**). Dilution of the extra-vesicular dispersion medium by water subjects the GUV to a hypotonic bath producing an osmotic differential (**B**), which renders initially flaccid vesicles stiff and replaces the initially optically uniform membrane surface by one characterized by a domain pattern of co-existing $L_d$ and $L_o$ phases at microscopic length scales. Solute is rendered as white particles, membrane, pink, and domain pattern in pink and purple. (**C**–**D**) The process in (**A**–**B**) exemplified by wide-field fluorescence (**C**) and deconvolved (**D**) images of a solution of GUVs consisting of POPC:SM:Ch (1:1:1) labeled with 0.5 mol% Rho-DPPE at 25°C containing 200 mM sucrose concentration, osmotically balanced by 200 mM glucose in (**C**), and under an osmotic differential of ~200 mM in (**D**) at 25°C. Scale bars: 15 μm.

The following figure supplement is available for figure 1:

**Figure supplement 1**. Undulating boundary of isotonic vesicles.

is used (*Figure 3*), we find that the oscillatory domain behavior is fully reproduced. Moreover, by preparing GUVs using gentle hydration (*Morales-Penningston et al., 2010*), we further confirm that the observed behavior is not adversely affected by the electroformation method (*Video 4*, *Figure 3—figure supplement 1*).

This oscillatory pattern of phase separation appears to be a cyclical isothermal phase transition resulting from oscillations in osmotic pressure and membrane tension, which characterize osmotic relaxation in vesicular compartments subject to osmotic differentials. A synergistic interplay of well-understood fundamental biophysical mechanisms—including selective membrane permeability for water (*Deamer and Bramhall, 1986*; *Peterlin and Arrigler, 2008*), osmotically-generated membrane tension, tension- and pressure-dependent membrane phase behavior (*Hamada et al., 2011*; *Portet et al., 2012*; *Uline et al., 2012*; *Givli et al., 2012*), and poration (*Sandre et al., 1999*; *Brochard-Wyart et al., 2000*; *Karatekin et al., 2003b*)—couple osmotic activity of water with spatial organization of membrane molecules (i.e., appearance of large, microscopic domains), such as those analyzed below.

The existence of an osmolyte concentration difference across the vesicular boundary triggers an osmotic relaxation process, which acts to reduce the pressure difference across the semi-permeable membrane by an influx of water (*Mui et al., 1993*). As water enters, the GUV swells ironing out the thermal

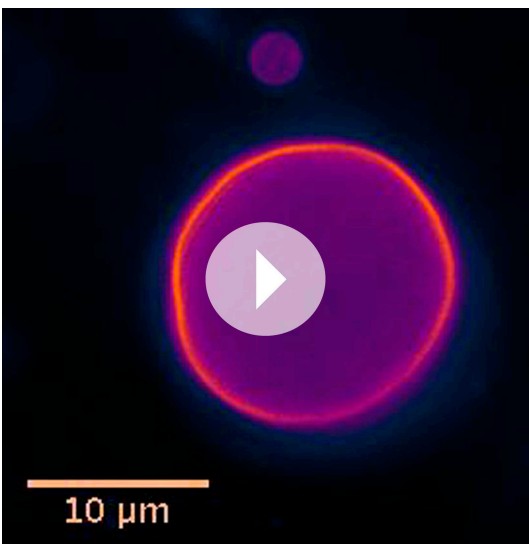

**Video 1**. Thermally excited undulations of isotonic GUVs. A video assembled from time-lapse fluorescence images revealing out-of-plane membrane fluctuations typical for non-tense GUVs in the absence of an osmotic gradient (50 vol% glycerol inside and outside). The osmotically balanced GUV consists of POPC:SM:Ch (1:1:1) labeled with 0.5 mol% Rho-DPPE (pseudo-colored magenta) and is imaged at 25°C.

undulations and rendering the vesicular boundary tense (*Figure 1A,C*) (*Haleva and Diamant, 2008*). At equilibrium, the lateral tension generated in the membrane compensates for the osmotic pressure, consistent with the law of Laplace, $\sigma$ (= $\Delta P$ $r/2$, where $\Delta P$ is the osmotic pressure difference and $r$, the vesicle radius).

A closer examination of the results above reveals that (1) the domains coarsen through collision and coalescence (*Figure 4A* and *Video 5*) and (2) the appearance of phase-separated state invariably coincides with the swollen, tense state of the GUV during the cyclical swell–burst processes (*Figure 4B,C*). Although both lateral tension and pressure difference influence membrane phase behavior in our osmotically driven case, it is instructive to consider how each of the two factors individually affects membrane phase behavior. A recent thermodynamic analysis and experiments examining the effects of mechanically generated tension reveal a lowering of miscibility phase transition temperature between the $L_o$ and $L_d$ phases with increase in tension ($dT/d\sigma$, ~−1 K [mNm⁻¹] ⁻¹) (*Portet et al., 2012*; *Uline et al., 2012*). However, how this shift in transition temperature affects membrane phase behavior and domain morphology is not obvious: a recent experimental study suggests that even tension alone can stabilize complex domain morphologies (*Chen and Santore, 2014*). The current and earlier observations in which osmotic differentials induce phase separation (*Hamada et al., 2011*) are clearly at variance with these predictions. An alternate explanation involves separate theoretical arguments, which require pre-existing phase separated domains in the optically homogeneous state. It suggests that the lateral tension elevates line tension between co-existing phases (*Akimov et al., 2007*). Therefore, although membrane tension disfavors nucleation of a new phase (by raising the energy barrier that must be met for the formation of critical nuclei), it can promote coalescence of small pre-existing nanoscale domains driven by minimization of line tension between $L_o$ and $L_d$ phase. Additional experiments using ternary lipid mixtures (DOPC, DPPC, and Chol), which have been thought not to produce nanodomains at temperatures above 20°C (*Hamada et al., 2011*), also produces oscillatory phase behavior (*Video 6*). This then suggests that the osmotically generated tension alone might be insufficient to explain the observed osmotically induced isothermal phase transition, and that the non-ideality in mixing is likely a consequence of a combined effect of the pressure and tension. Indeed, a theoretical model by Givli and Bhattacharya (*Givli et al., 2012*), explicitly introducing osmotic pressure contributions within the generalized Helfrich energy treatment, suggests that pressure can perturb isothermal phase diagram, driving domain formation primarily by affecting the interaction between geometry and composition.

This tension and pressure-mediated appearance of the phase-separated state in osmotically swollen membranes, however, does not account for the oscillations in domain pattern: (1) why does the osmotically swollen vesicle characterized by large microscopic domains return to a homogeneous state, and (2) what prompts subsequent cycles of phase separation? A closer look at the temporal dynamics reveals that the process does not persist indefinitely. The period of oscillation between optically uniform and phase-separated states increases with the passage of time (*Figure 4D–F*). The cycle period—defined as the time elapsed between two consecutive instances of homogeneous fluorescence—increases three to 10-fold, before reaching a non-oscillating quiescent state, 60–120 min after the imposition of the osmotic differential. This 'fatigue' in the oscillatory phase separation process suggests that the driving force (i.e., the osmotic differential and accompanying tension) must weaken with each cycle, which requires a separate mechanism for solute efflux.

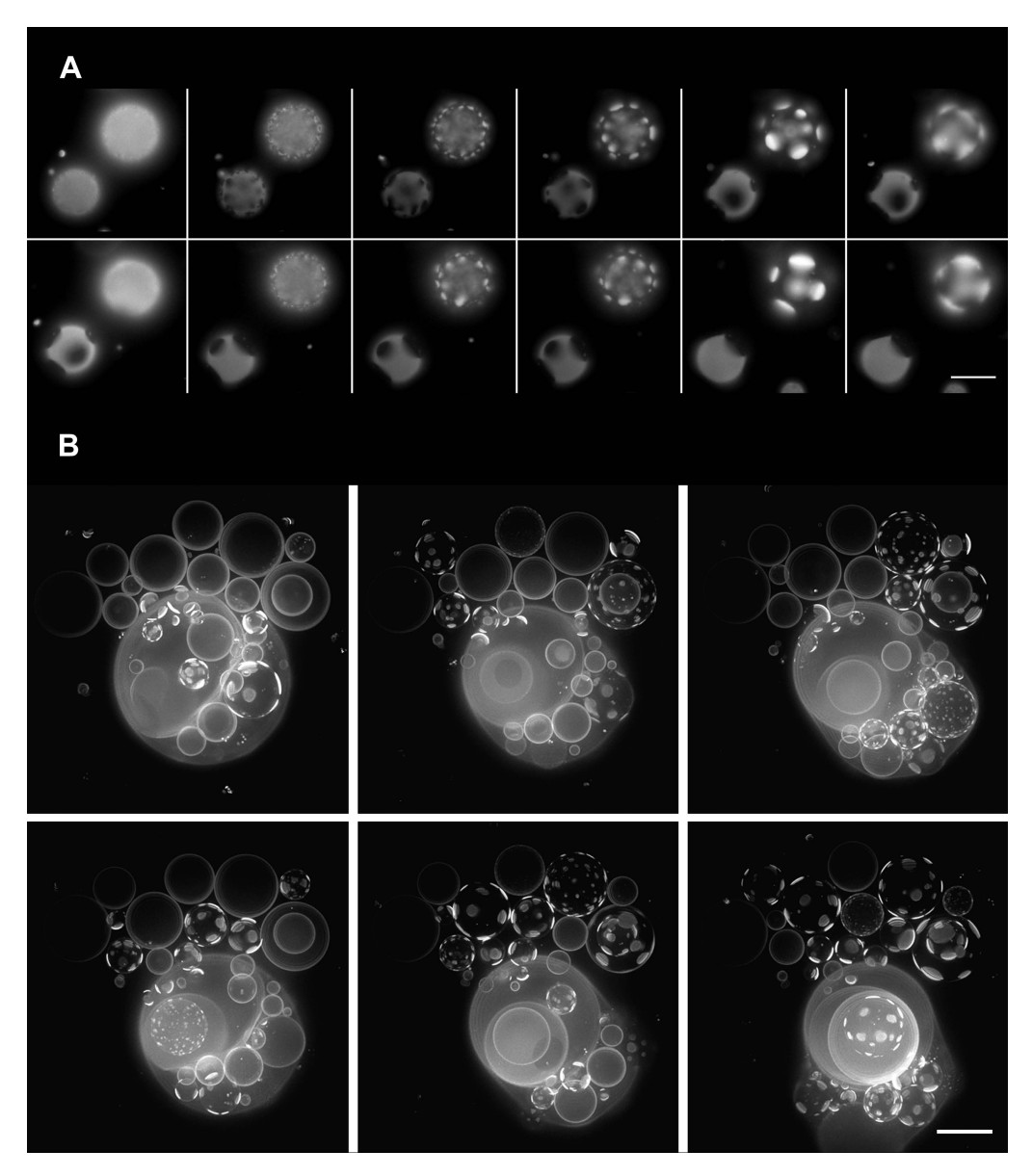

**Figure 2**. Oscillatory phase separation in hypertonic giant unilamellar vesicles subject to an osmotic imbalance. (**A**), Selected frames from a video of time-lapse fluorescence images (**Video 2**) illustrating stages of domain dynamics during two consecutive cycles of oscillatory phase separation (t = 0 s, 9 s, 12 s, 15 s, 25 s, 27 s, 29 s, 188 s, 191 s, 193 s, 246 s, and 247 s). The GUVs imaged consist of POPC:SM:Ch (1:1:1) labeled with 0.5% Rho-DPPE, encapsulating 1 M sucrose, diluted in deionized water, at room temperature. Scale bar: 10 μm. (**B**) Selected images from time-lapse fluorescence images (**Video 3**) showing asynchronous cycling in a population of GUVs (t = 0 s, 98 s, 148 s, 294 s, and 448 s). The images are projections of Z-stacks of the lower hemispheres of GUVs consisting of POPC:SM:Ch (1:1:1) labeled with 0.5 mol% Rho-DPPE, encapsulating 200 mM sucrose, diluted in deionized water at 25°C (n = 5). Scale bar: 15 μm.

## Membrane poration in osmotically tense vesicles

It is known that membrane lysis proceeds via cascades of pores during each cycle of the swell–burst sequence (**Karatekin et al., 2003b**). This strikingly regular, temporal cascade of pores is fully reproduced in our case (**Figure 5**, **Figure 5—figure supplement 1**): during the swell segment of each oscillation cycle, a single microscopic pore, several micrometers across, becomes visible under conditions of maximum swelling and largest domain size, typically for a period not exceeding 1.0 s

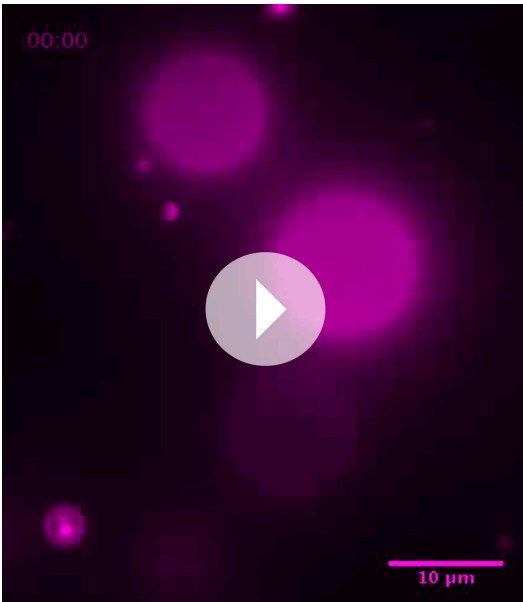

**Video 2**. Oscillatory domain dynamics in GUVs immersed in hypotonic bath. Time-lapse images of a bottom view of GUVs consisting of POPC:SM:Ch (1:1:1) labeled with 0.5% Rho-DPPE (pseudo-colored magenta) under a net osmotic differential. The GUVs encapsulate 1 M sucrose in their interior, and the external dispersion medium is MilliQ water. A striking temporal pattern of oscillatory phase separation revealing appearance, coalescence, and dispersion of optically resolved domains is evident (see manuscript for details).

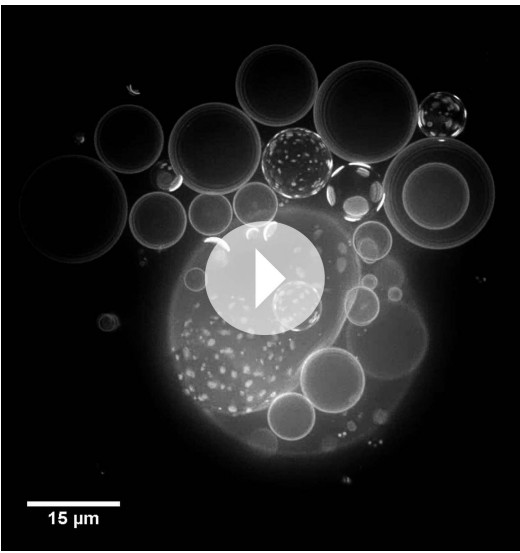

**Video 3**. Domain dynamics of GUVs in hypotonic bath. Video assembled from time-lapse images of Z-stack projections of the bottom hemispheres of GUVs, consisting of POPC:SM:Ch (1:1:1) labeled with 0.5 mol% Rho-DPPE (white). The vesicles encapsulated 200 mM sucrose and were diluted in MilliQ water at 25°C.

(*Video 7*). According to classical nucleation theory, the cost (*E*) of creating a pore in a tense membrane is determined by the competition between membrane tensional energy ($-\pi r^2 \sigma$) and the line tension energy ($+2\pi r\gamma$) at the edge of the pore. Thus, under conditions of sufficient membrane tension ($dE/dr > 0$), pores nucleate and grow, enabling solute efflux (*Sandre et al., 1999*; *Peterlin and Arrigler, 2008*). Although domain formation is not required for pore-formation, the probability of pore-nucleation might be enhanced by surface defects, such as are present at the boundary between co-existing phases, since the energy required to open a pore (>40 $K_BT$) is considerably higher than the thermal activation energy (*Karatekin et al., 2003b*). The long life spans (~1 s) of the pores are likely supported by two opposing processes, namely osmotic influx of water and the leakage rate of solute through the pore (*Koslov and Markin, 1984*). Subsequent healing of the pore is promoted by the reduction in the net membrane area and partial loss of the encapsulated solutes, both of which reduce membrane tension, $\sigma$ (*Karatekin et al., 2003b*). Thus, during each membrane rupture event, only a fraction of the intravesicular solute is released before the bilayer reseals, leaving the vesicle hyperosmotic, albeit with a reduced osmotic differential. This then prompts subsequent cycles of water influx, vesicle swelling, and rupture until sufficient intravesicular solute is lost and the Laplace tension in the membrane is able to compensate for the residual osmotic pressure (*Ertel et al., 1993*).

The oscillatory phase separation above does not require isolated vesicles but becomes integrated with other known shape transformations in GUVs experiencing tension (*Seifert, 1997*). Using structurally complex GUVs, which hierarchically embed smaller ones with different osmolyte concentrations, we found that the domain dynamics becomes coordinated with the previously well-known process of expulsion of internal 'organelle' vesicles (*Video 8* and *Figure 6*) (*Moroz et al., 1997*; *Oglecka et al., 2012*). The observations above further show how local inhomogeneity in the distribution of solute, namely sucrose in the present case, in nested or hierarchical vesicular compartments in single solutions can produce localized oscillatory phase behavior in component vesicles. These observations also suggest that the oscillatory phase separation can be regarded as a type of amplified mechanosensor for solute concentration differences and osmotic differentials.

The emergence of oscillatory domain dynamics appears to be a well-coordinated membrane response to osmotic stress through an isothermal

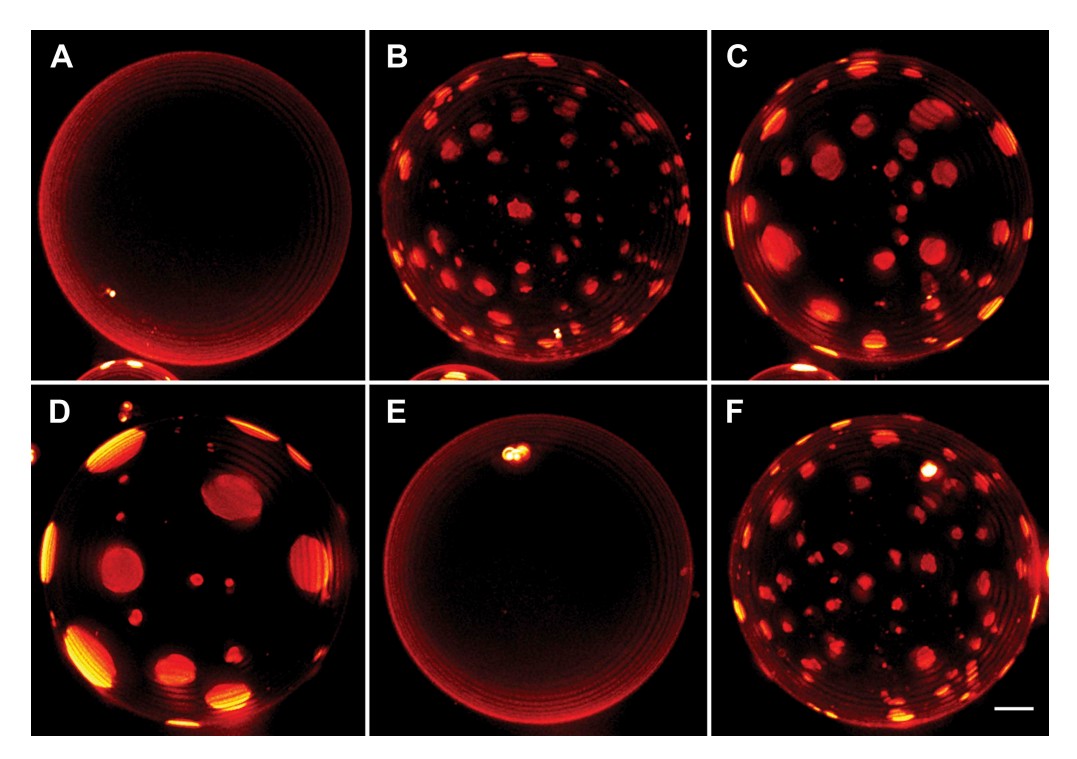

**Figure 3**. Interrupted imaging of oscillatory phase separation. Z-stack projections of height-resolved fluorescence images of the lower hemisphere of a GUV consisting of POPC:SM:Ch (1:1:1) labeled with 0.5% Rho-DPPE (pseudo-colored red). The GUV encapsulates 200 mM sucrose, and the external dispersion medium is diluted in MilliQ water. Images are acquired at 25°C at arbitrary time points; (**A**) 0 s, (**B**) 99 s, (**C**) 148 s, (**D**) 299 s, (**E**) 550 s, and (**F**) 692 s. The first image was taken ~2 hr after imposing the osmotic gradient. Scale bar: 5 μm.
The following figure supplement is available for figure 3:

**Figure supplement 1**. Oscillatory phase separation in complex GUVs prepared by hydration.

phase transition resulting from a highly coordinated interplay between elementary physical mechanisms (*Figure 7*). Specifically, the processes of (1) osmotically triggered water influx; (2) retention of osmotic pressure and build-up of membrane tension ironing out thermal undulations; (3) appearance of microscopic domains in the membrane subject to osmotic pressure and lateral tension; (4) coarsening of domains; (5) appearance of a short-lived transient pore, which enable partial solute efflux reducing osmotic pressure and membrane tension; and (6) consequent pore-closure resulting in closed GUVs with reduced osmotic differential—repeat until the sub-lytic osmotic pressure is reached. Although individual biophysical processes leading to the oscillatory domain dynamics during this osmotic relaxation process are well-appreciated, the observations reported here bring to focus several features of vesicle behavior, which are best poorly appreciated. *First*, the seemingly autonomous vesicle response—in which an external osmotic perturbation is managed by a coordinated and cyclical sequence of physical mechanisms allowing vesicles to sense (by domain formation) and regulate (by solute efflux) their local environment–suggests a primitive form of a quasi-homeostatic regulation in a synthetic material system (*He et al., 2012*), that is, a simple microemulsion produced from simple components, namely, lipids, water, and osmolytes. *Second,* these observations illustrate how out-of-plane osmotic activity of water becomes coupled with membrane's in-plane compositional degrees of freedom producing an exquisite and complex response. It underscores the intrinsic coupling between membrane phase and mechanical tension. *Third*, by highlighting the complexity of lipid vesicles, these results offer an important *caveat* in implementing giant vesicles as experimental models in scenarios where osmotic imbalances can dominate vesicle response.

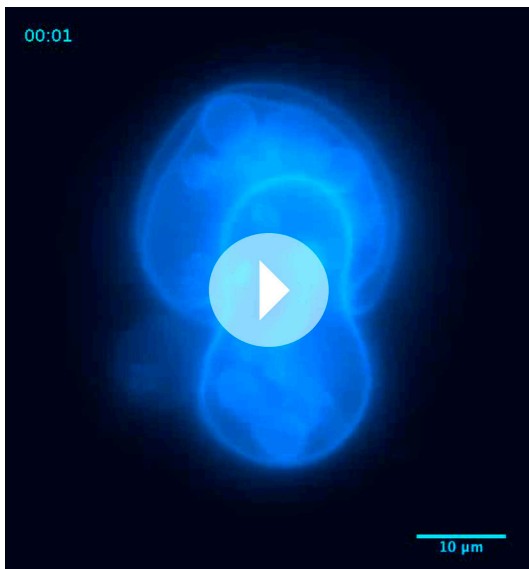

**Video 4**. GUVs prepared by gentle hydration reproduce the oscillatory domain dynamics. GUVs prepared by 'gentle hydration' consisting of POPC:SM:Ch (1:1:1) labeled with 0.5 mol% Rho-DPPE (pseudo-colored blue), encapsulating 200 mM sucrose. Hypotonic conditions are established by dilution in deionized water at 25°C. Vesicles exhibit osmotic swelling and oscillatory domain dynamics comparable to that seen for electroformed GUVs under the same conditions.

The findings reported here—illustrating a complex dynamic relationship between the membrane's compositional degrees of freedom with external osmotic imbalance—might be biologically relevant. While large microscopic domains of co-existing liquid phases are thought to not exist in many living cells, a recent study reveals such domain texture in yeast vacuole membranes (*Toulmay and Prinz, 2013*). In this study, under conditions of nutrient deprivation, pH changes, and changes in the growth medium have been shown to segregate into microscopic domains in what appears to be a sterol-dependent manner, reminiscent of synthetic giant vesicles. Since one of the functions of yeast vacuoles is to regulate osmotic pressure, it seems tempting to consider whether vacuolar domains also undergo large-scale domain reorganization under osmotic stimuli and contribute to the physiological function. The formation (and dissolution) of compositionally differentiated membrane domains (e.g., lipid rafts) is often associated with alterations in the conformations (and activity) of many signaling proteins (*Simons and Toomre, 2000*), which partition within them. It appears plausible that the domain reorganization stimulated by the osmotic activity of water might provide the cell a generic mechanism to respond to the physical perturbation, such as by activating mechanosensitive ion-channels and serving as sensors for signaling and stress transmission (*DuFort et al., 2011*; *Stamenovic and Wang, 2011*; *Wood, 2011*). *Fourth*, although fatty acid based prebiotic amphiphiles exhibit different mechanical properties (e.g., elastic properties and permeability characteristics) compared to phospholipids, it appears likely that amphiphilic osmoregulation, such as we witness, might have given a thermodynamic advantage to early protein-free protocells (*Hanczyc et al., 2003*; *Chen et al., 2004*; *Oglecka et al., 2012*) to survive (and even utilize) drastic environmental osmotic shifts.

## Key biophysical considerations

Elementary physical mechanisms of membrane permeability (*Fettiplace and Haydon, 1980*; *Deamer and Bramhall, 1986*; *Rawicz et al., 2008*), osmotic swelling (*Taupin et al., 1975*; *Mui et al., 1993*; *Haleva and Diamant, 2008*; *Peterlin and Arrigler, 2008*; *Peterlin et al., 2012*), tension dependence of lateral phase separation (*Hamada et al., 2011*; *Portet et al., 2012*; *Uline et al., 2012*), pressure-dependent membrane phase separation (*Givli et al., 2012*), and membrane poration (*Needham and Hochmuth, 1989*; *Zhelev and Needham, 1993*; *Sandre et al., 1999*; *Brochard-Wyart et al., 2000*; *Karatekin et al., 2003a*; *Karatekin et al., 2003b*; *Levin and Idiart, 2004*; *Farago and Santangelo, 2005*; *Riske and Dimova, 2005*; *Evans and Smith, 2011*) have all been extensively studied and well-documented in the existing literature. Below, we recapitulate the key aspects of these mechanisms (and obtain rough estimates for key observables), which constitute key parts of the emergent behavior reported herein.

## Osmotic relaxation across semipermeable media

Consider a case in which dilution of the external dispersion medium results in the creation of an initial osmotic differential of 200 mM (sucrose). Applying van't Hoff's equation ($\Delta P_{osm} = RT\Delta c$, where $R$ is the gas constant and $T$ the absolute temperature), this concentration gradient corresponds to an excess intravesicular osmotic pressure of 0.5 MPa. As a result of the pressure difference, an osmotic relaxation process sets in. Because of large differences in permeability of water ($\sim 10^{-3}$–$10^{-4}$ cm s$^{-1}$) and sucrose ($\sim 10^{-8}$ cm s$^{-1}$), however, the relaxation process is determined by the

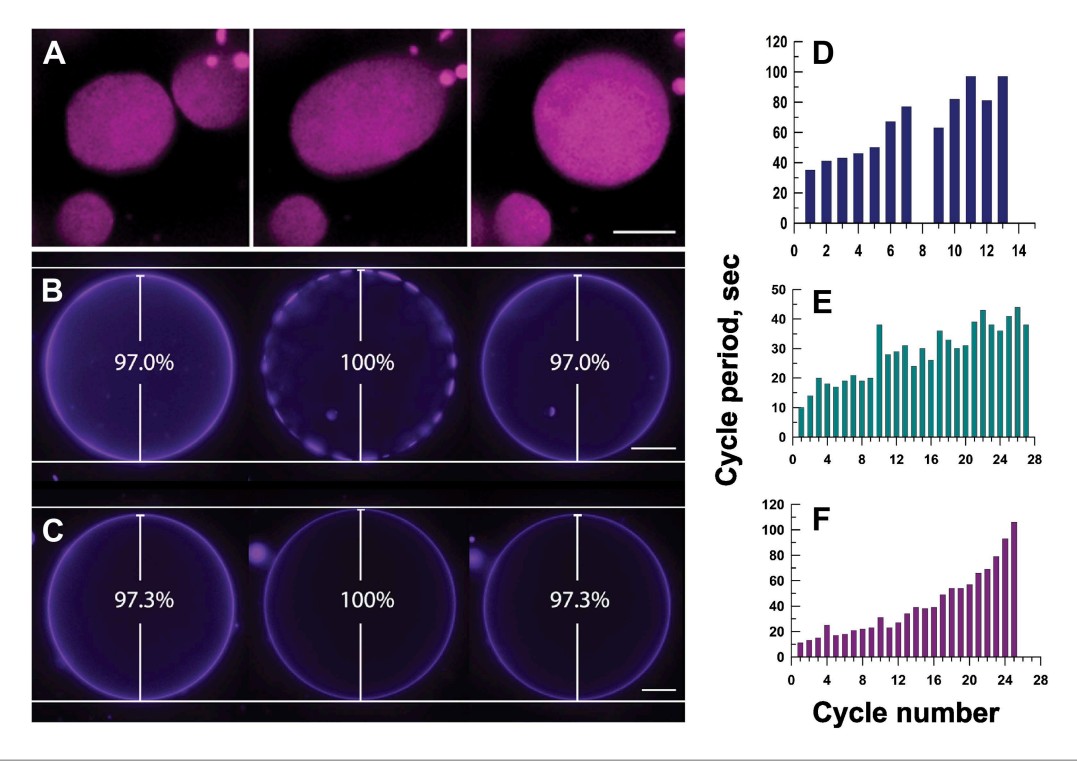

**Figure 4**. Mechanisms responsible for oscillatory phase separation in GUVs subject to osmotic differentials. (**A**) *Domain coarsening*. Selected frames from *Video 4* illustrating collision and coalescence of domains during a swell segment of the GUV oscillations ($L_d$ phase, pseudo-colored magenta). Images are 1 s apart focused on a region of interest located at the bottom of a GUV. Scale bar: 5 μm. (**B–C**) *Relationship between vesicle swelling and phase-separation*. Fluorescence images revealing (**B**) that largest domains are observed under conditions of maximal swelling (t = 0 s, 8 s, and 106 s). Scale bar: 10 μm. (**C**) Control experiment using single component POPC GUVs, labeled with 0.5% Rho-DPPE, encapsulating 200 mM sucrose, diluted in deionized water at 25°C, confirm that the GUV swelling does not require domain formation and/or reorganization. Scale bar: 10 μm. (**D–F**) *Increase of cycle period during oscillatory domain dynamics*. A bar chart showing successively increasing periods of domain growth/dispersion cycles in GUVs (**D**) 42.0 μm, (**E**) 26.3 μm, and (**F**) 10.7 μm in diameter. A cycle period is defined as the time elapsed between two consecutive instances of appearance of uniform fluorescence. Except for control in (**C**), all data were collected using POPC:SM:Ch (1:1:1) GUVs, labeled with 0.5% Rho-DPPE, encapsulating 200 mM sucrose, diluted in deionized water at 25°C.

significant differences in time scales for permeation of water and the solute: rapid water permeation governs the response of the solute-encapsulating hypertonic GUV by rapidly adjusting its volume and reducing the effective pressure difference across the membrane (*Haleva and Diamant, 2008*; *Peterlin et al., 2012*).

The residual osmotic pressure then necessarily generates a lateral tension in the membrane, which compensates for the internal fluid force. Following the law of Laplace ($\sigma = \Delta P\, r/2$), the imposed osmotic pressure of 0.5 MPa translates into the applied membrane tension, $\sigma$, of 2.5 N m$^{-1}$ for a vesicle, 10 μm in radius—approximately three orders of magnitude larger than necessary for membrane lysis (~3–5 mN m$^{-1}$) (*Portet and Dimova, 2010*).

It is instructive to note, however, that the actual tension that develops in the membrane is much lower. GUVs prepared by electroformation invariably display large variations in size, shape, and area to volume ratio. As a result, upon immersion in hypotonic solution, osmotic influx of water first transforms the initial non-spherical shapes into spherical ones (*Hamada et al., 2011*). The drop in the osmolyte concentration during this transformation effectively reduces the osmotic pressure difference, which the vesicle experiences, compared to the applied one. Note also that this reduction in osmotic pressure during the initial shape transformation is different for different vesicles within single populations.

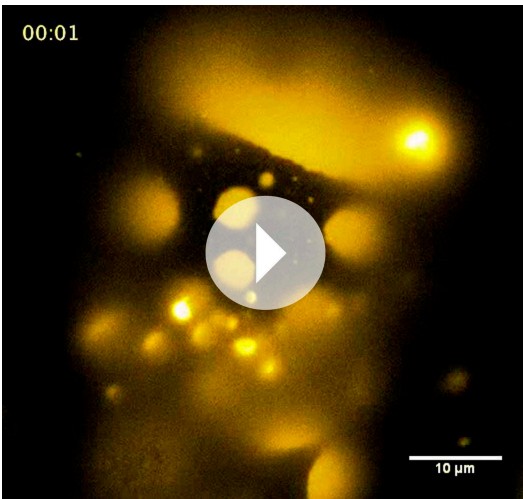

**Video 5**. Evidence for domain merger by collision and coalescence. Time-lapse wide-field fluorescence images of the lower hemisphere of POPC:SM:Ch (1:1:1) GUVs labeled with 0.5 mol% Rho-DPPE (pseudo-colored yellow), encapsulating 200 mM sucrose, diluted in MilliQ water at 23°C. Domain–domain coalescence, followed by line-tension driven shape transformations, drives domain growth. Frames are collected at 1 s intervals.

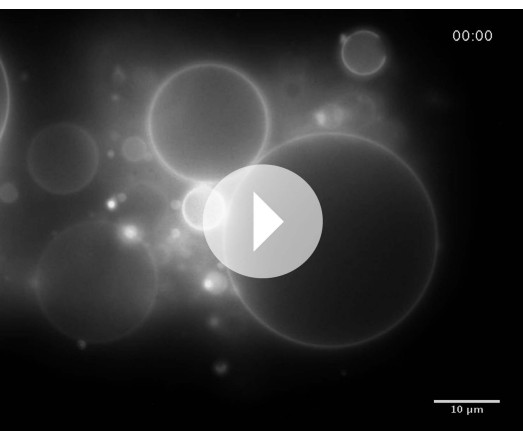

**Video 6**. Oscillatory domain dynamics in mixed ternary system known to exist in single liquid state in the absence of net osmotic differential. Time-lapse wide-field fluorescence images of DOPC:DPPC:Ch (5:2:3) GUVs labeled with 0.5 mol% Rho-DPPE (pseudo-colored yellow), encapsulating 200 mM sucrose, diluted in MilliQ water at 23°C. Frames are collected at 1 s intervals.

Thus, although quantifying actual membrane tension in our experiments is difficult, the observations of tense membranes and subsequent poration indicate that the osmotic perturbation is sufficient for most GUVs to surpass their maximum volume limit, developing appreciable membrane tension and pressure differences.

## Development of membrane tension in hypotonic media

GUV membranes in isotonic media are flaccid (*Figure 1A,C*). The actual membrane area, $A$, is higher than the projected area $A_p$. The excess area, $\Delta A$ ($=A - A_p$), ensures that the flaccid GUV is essentially free of tension ($\sigma$) and exhibits thermally-excited undulations (*Seifert, 1997*). In a hypotonic bath, the GUV assumes a spherical shape and the membrane fluctuations become suppressed (*Figure 1B,D*), dilating the vesicular volume by >15%. During this transformation, $A_p$ also increases proportionately. In the low-tension regime, where the shape fluctuations are not completely ironed out, $A_p$ increases logarithmically with $\tilde{\sigma}$. This is followed by a high tension regime, in which $A_p$ climbs linearly with $\sigma$ because of the stretching of the molecular areas. The area dilation ($\alpha$) in the membrane is given by the superposition of an area increase due to reduction of membrane undulations and expansion in area per molecule (*Evans and Rawicz, 1990*).

$$\alpha = \frac{kT}{8\pi\kappa_c}\left[\ln\left(1 + c\sigma\kappa_c\right) + \frac{\sigma}{E}\right] \tag{1}$$

where $\kappa$ and $E$ are the elastic moduli for bending and area expansion, respectively, and the coefficient $c$ for tension is $1/24\pi$. The cross-over between the two regimes occurs at the critical tension of $E$ ($=kT/8\pi\kappa_c$), which for a typical phospholipid membrane falls between 0.1–1 mN m$^{-1}$. Because of their large dimensions, GUVs develop appreciably greater membrane tensions (>> 1 mN m$^{-1}$) under even small osmotic gradients (mM range), placing our experiments in the high-tension regime. This is also confirmed by the routine formation of pores evident in our data (see above).

## Pore-induced relaxation of membrane tension

(*Sandre et al., 1999*; *Brochard-Wyart et al., 2000*; *Karatekin et al., 2003a*; *Karatekin et al., 2003b*) An increase in the projected area ($A_p$) of the membrane in the presence of tension can be expressed in terms of a hypothetical radius, $R_o$, which the vesicle would adopt were its membrane tension absent ($\sigma = 0$).

$$4\pi R_0^2 = 4\pi R_0^2\left[1 + \frac{\sigma_0}{E}\right] \tag{2}$$

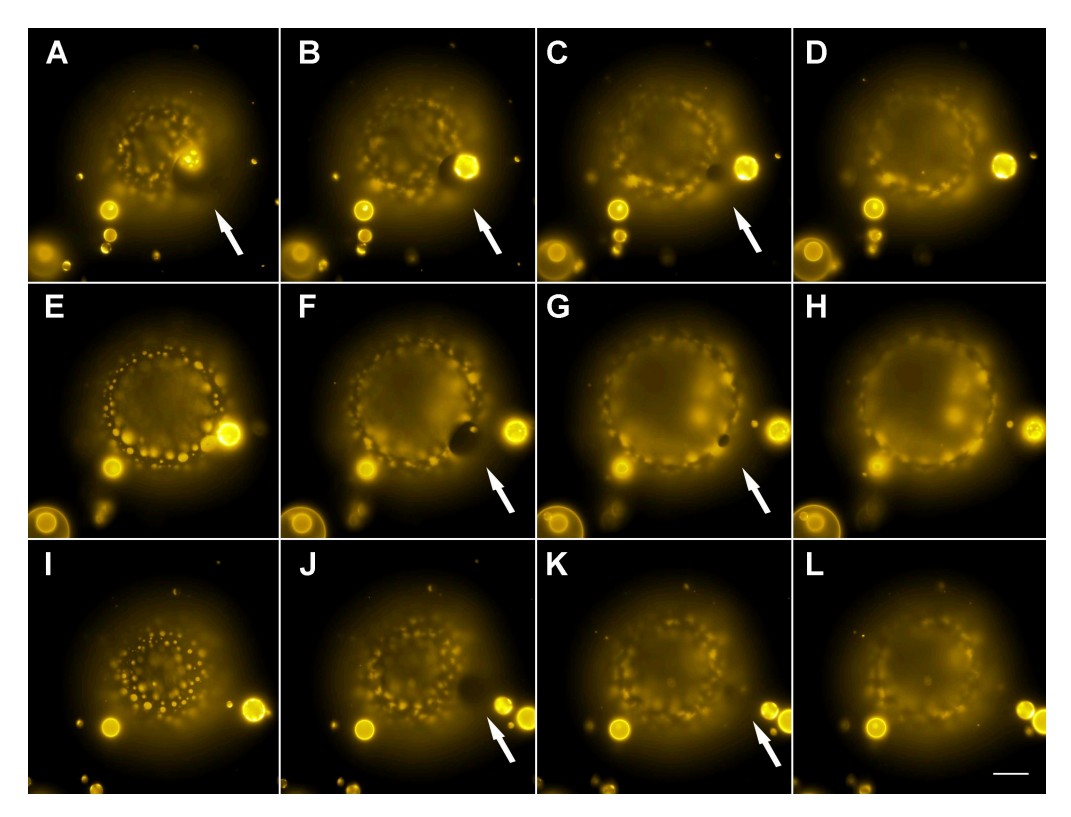

**Figure 5**. Evidence for the formation of microscopic pores during each individual cycle of oscillatory phase separation. (**A–L**) Wide-field fluorescence images of microscopic pore formation (~5–15 µm in diameter; indicated by arrows) observed in three consecutive swell–burst cycles of a phase-separating GUV. A single pore appears during each phase separation cycle and reseals within 1 s. The POPC:SM:Ch (1:1:1) GUV labeled with 0.5% Rho-DPPE (pseudo-colored yellow), encapsulates 200 mM sucrose, and is immersed in deionized water at 25°C. Images collected 20 min after imposition of the osmotic differential. Height-resolved (increment, 0.5 µm) images shown at arbitrary time intervals following the first frame. (**A–L**) 0 s, 0.3 s, 0.6 s, 0.9 s, 9.3 s, 9.6 s, 9.9 s, 10.1 s, 15.3 s, 15.6 s, 15.9 s, and 16.1 s. Scale bar: 15 µm. Cascades of pores have been observed more than five times.

The following figure supplement is available for figure 5:

**Figure supplement 1**. Evidence for pore-formation.

---

After a pore opens, the tension in the membrane, $\sigma_0$, drops to $\sigma$. Thus,

$$4\pi R_0^2\left[1+\frac{\sigma_0}{E}\right] = 4\pi R_0^2\left[1+\frac{\sigma}{E}\right] + \pi r^2 \tag{3}$$

which yields an estimate for the critical radius to which the pore must grow to relax the membrane tension completely.

$$r_c = 2R_0\left(\frac{\sigma_0}{E}\right)^{1/2} \tag{4}$$

Rearranging the *Equation (3)* in terms of critical radius, the stress equation can be written in terms of a vesicle's geometric parameters (under conditions of no leakage).

$$\frac{\sigma}{\sigma_0} = 1 - \frac{r^2}{r_c^2} - \frac{4\left(R_i^2 - R^2\right)}{r_c^2} \tag{5}$$

This equation describes two conditions for tension–relaxation following pore opening. *First*, as the pore grows, the first negative term in the equation above increases, reducing membrane tension consistent with

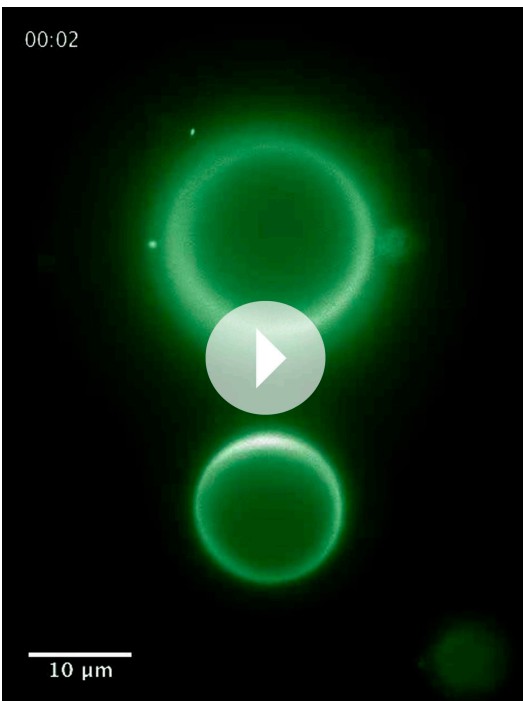

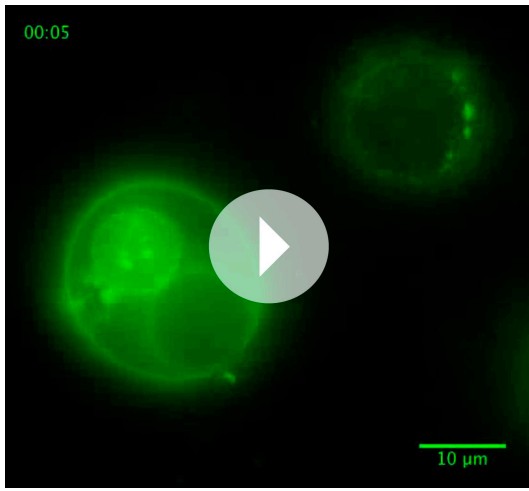

**Video 7**. Evidence for pore formation. Equatorial view of a POPC:SM:Ch (1:1:1) labeled with 0.5 mol% Rho-DPPE (pseudo-colored green), encapsulating 50vol % glycerol, diluted in MilliQ water at 25°C. Pore formation can be clearly seen at 1.1 s, just prior to the disappearance of domains and size shrinkage of the GUV.

**Video 8**. Oscillatory phase separation during expulsion of daughter GUVs. A video assembled from time-lapse fluorescence images of phase-separating GUVs containing internal 'organelle' vesicles. GUVs consisting of POPC:SM:Ch (2:2:1) labeled with 0.5 mol% Sphingomyelin-Atto647N (SM-647N) (pseudo-colored green), encapsulate 1 M sucrose, and diluted in MilliQ water at 25°C. The video reveals shifting patterns of osmotic pressure and tension during expulsion of the internal vesicles after an osmotic differential had been established. Key steps include (A) a homogeneous, flaccid mother vesicle encapsulating tense daughter vesicles, at a time point prior to vesicle expulsion; (B) just after expulsion, the daughter GUV remains tense exhibiting oscillatory phase separation, while the mother GUV is left deflated and homogenous due to the sudden loss of volume; (C) The mother GUV subsequently becomes inflated by influx of water; and (D) the mother GUV begins to exhibit oscillatory phase separation.

the physical picture that pore opening causes lipids to distribute over a smaller area, thus reducing tension. *Second*, the efflux of the vesicular content following the opening of the pore reduces R, making the second negative term larger, reducing tension ($\sigma$). Together, they set the stage for pore closure.

## Solute leakage and pore lifetimes

Solute efflux through an open pore in an osmotically stretched membrane occurs under a complex hydrodynamic scenario. Pore radius, vesicle volume, and osmotic pressure differences all change with time, and persistent excess solute concentration maintains conditions for water influx, all of which influence shear stresses associated with the net outward flow. Below, we consider the effusion mechanism following Levin and Idiart (*Idiart and Levin, 2004*; *Levin and Idiart, 2004*).

A simple diffusion analysis, such as summarized below, provides a comparison between the amount of solute released in each cycle for the experimental life-time of microscopic pores, which we witness.

Diffusive current through a pore of size, *r*, is given by

$$j = \pi r^2 c \frac{D}{R}, \tag{6}$$

where *c* is the sucrose concentration, *D* is the solute diffusivity, and *R* represents the vesicle radius.

Comparing the diffusive current with the rate of drop of sucrose concentration within the GUV, then yields,

$$\frac{4}{3}\pi R^3 \frac{dc}{dt} = -j \tag{7}$$

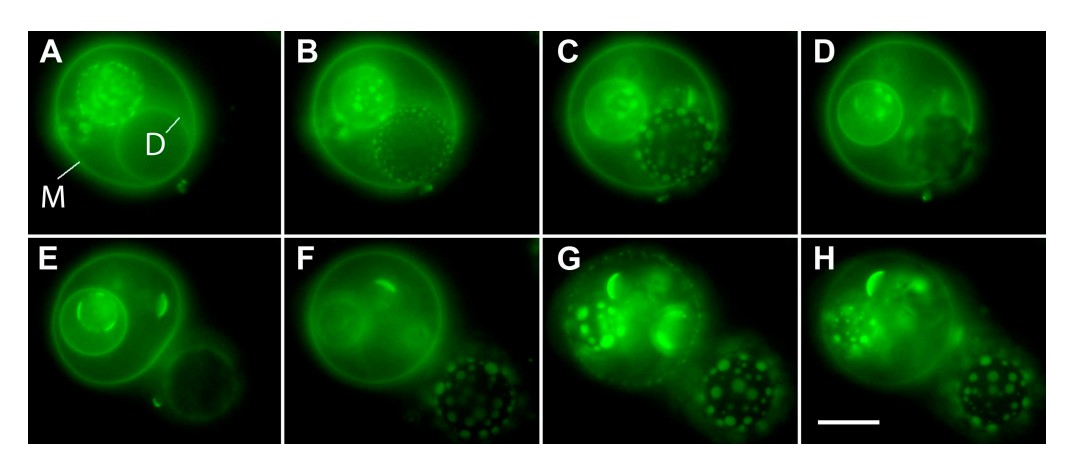

**Figure 6**. Osmotic gradients sensed by the membrane and visualized by oscillatory phase separation in nested vesosomes. Selected frames from **Video 7** showing hierarchical membrane structures of POPC:SM:Ch (2:2:1) GUVs labeled with 0.5% SM-Atto647N (pseudo-colored green), encapsulating 1 M sucrose, submerged in MilliQ water at 25°C. In panel (**A**), we define the entrapping mother vesicle as *M* and daughter vesicle of interest as *D*. Both *M* and *D* initially exhibit homogenous fluorescence from their membranes, but store different amounts of tension (*M* is flaccid, while *D* appears tense). (**B**) The homogeneous fluorescence from *D* is replaced by the appearance of optically resolved domains. In the meantime, *M* becomes more spherical. (**C**) The domains of *D* have increased in size, and *M* has now reached an almost spherical shape. (**D**) Expulsion of the tense *D* vesicle. This image acquired during a transient pore formation suggests that the intravesicular pressure and/or crowding is reduced via preferential expulsion of daughter. This event, we surmise, also delays the onset of domain formation by reducing the swelling of the *M* vesicle. (**E**) *M* is returned to a flaccid state, remaining homogenously fluorescent, consistent with the reduction in swelling and a reduction of osmotic pressure. At the same time, *D* experiencing a new hypotonic medium gets engaged in swell–burst cycles. (**F**) Further inflation of GUVs leads to *M* adopting a tense spherical configuration, while yet retaining homogenously fluorescent state, while *D*'s domain sizes continue to grow. (**G**) The continued swelling of *M* finally leads to phase separation. (**H**) Domains in *M* disappear producing homogeneous state, consistent with the oscillatory phase separation under osmotically generated tension. Panels correspond to (**A**–**H**) 0 s, 6 s, 14 s, 18 s, 20 s, 62 s, 103 s, and 118 s. Scale bar: 10 µm.

Solving the differential **Equation (7)** above, we find that the concentration decay adopts an exponential profile,

$$c = c_0 e^{-t/\tau} \tag{8}$$

where the characteristic effusion time $\tau$ is

$$\tau = \frac{4R^2}{3r^2 D} \tag{9}$$

Using $D = 10^{-9}$ m²/s for sucrose in water and pore size, $r = 5$ µm, we find that for a GUV of radius 10 µm, the effusion time is ~0.5 s, comparable to our experimental estimate for the lifetime of pores (<1 s). This then suggests that pore lifetimes are sufficient to allow partial solute leakage (fractional loss, ~1/e) required to relax membrane tension and promote pore closure through effusion alone per cycle. Although the actual dynamics of solvent and solute transport across osmotically imbalanced vesicles are likely to be much more complex, the model above provides approximate estimates for the expected values.

## Materials and methods

### Materials

Sphingomyelin (chicken egg) and cholesterol were purchased from Carbosynth, Berkshire, UK. POPC (egg) (1-palmitoyl-2-oleoyl-*sn*-glycero-3-phosphocholine) and Rhodamine-DPPE (lissamine rhodamine

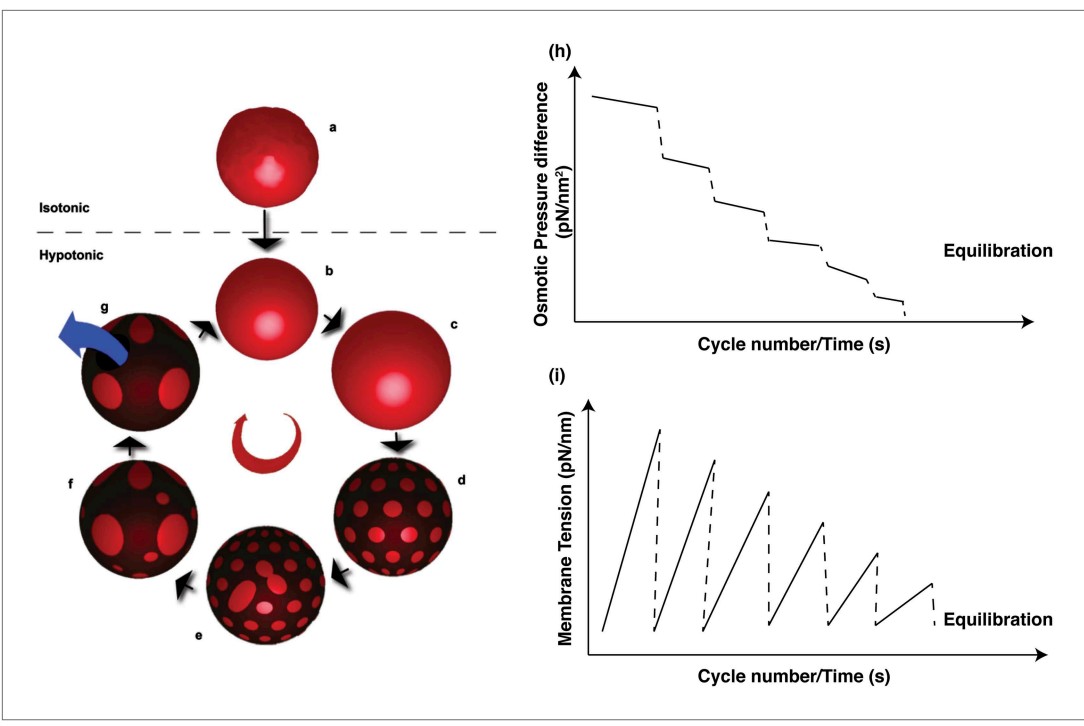

**Figure 7**. Schematic representations of physical mechanisms and changes in membrane properties during vesicular osmoregulation. (Left panel) (**A**) GUV in isotonic medium exhibiting a flaccid morphology. (**B**–**C**) Immersion in a hypotonic bath initiates an osmotically triggered influx of water rendering the GUV tense. (**D**–**F**) The optically uniform vesicular surface breaks up into a pattern of microscopic domains, which grow by collision and coalescence. (**G**) Transient appearance of a microscopic pore (~0.3–0.5 s lifetime), enabling solute efflux and tension relaxation, which drives pore closure, producing closed GUVs with a reduced osmotic differential and homogenous surface. Steps (**B**–**G**) repeat until the sub-lytic solute concentration differential is reached and the Laplace tension in the membrane is able to compensate for the residual osmotic pressure. (Right panel) Temporal cascades of osmotic pressure (**H**) and oscillations in membrane tension (**I**) during osmotic relaxation of giant vesicles subject to hypotonic bath. Note that the relative rates implied in the schematic are only best-guess estimates.

B 1,2-dihexadecanoyl-*sn*-glycero-3-phosphoethanolamine, triethylammonium salt), (also abbreviated Rhodamine-DHPE) were acquired from Avanti Polar Lipids, Alabama, USA. Sphingomyelin-Atto647N (SM-647N) was from Atto-Tec, Germany. Sucrose and glucose were from USB Corporation, Cleveland, OH, USA.

## Electroformation of giant unilamellar vesicles

Appropriate amounts of chloroform solutions of desired lipid mixtures doped with a small concentration of lipid-conjugated fluorescent dye were deposited onto clean ITO-coated glass surfaces within the area delimited by a small O-ring, and allowed to dry. Subsequently, the resulting dried lipid cake—containing ~60 µg of lipids and 0.5 mol% lipid-conjugated dye—was hydrated with 300 µl sugar solution of choice, flooding the O-ring enclosed area to the rim. The hydrated sample was then carefully covered by placing a second ITO-coated glass slide, avoiding entrapment of air bubbles. Electroformation (*Angelova et al., 1992*) was carried out at 45°C, above the gel–fluid transition temperatures of the lipid mixtures, using a commercial Vesicle Prep Pro (Nanion, Munich, Germany) chamber. Application of an AC current at 5 Hz and 3 V for 120 min yielded high abundance of 5–50 µm sized GUVs with excellent reproducibility.

## Gentle hydration method for giant unilamellar vesicle preparation

Same as electroformation above, except that no electrical current was applied (*Morales-Penningston et al., 2010*).

## Wide-field deconvolution microscopy

A DeltaVision microscope (Applied Precision, Inc., Washington, USA), fitted with a PLAPON 60XO/1.42 NA oil-immersion objective from Olympus and DAPI, TRITC, FITC, and CY5 Semrock filters (New York, USA), was used for imaging of GUVs in real-time using wide-field deconvolution fluorescence microscopy. Samples were imaged in 8-well chambers fitted with coverslip bottoms (Nunc, Rochester, USA). The 8-well chamber was fitted inside a custom made housing attached to a heating/cooling system, which also was designed to regulate the temperature of the objective. This assured that temperature differences between the sample and the lens would be kept to a minimum, and thus potential convective water flow inside the sample avoided. The temperature was monitored using a thermostat that was submerged into the sample volume. Briefly, 5 µl of sugar-encapsulating GUVs were placed inside a well and gently diluted in 200 µl of deionized water (18 megaohm cm). The osmolyte-loaded GUVs subsequently settled to the bottom of the coverslip. Osmotic differentials were generated in all experiments by diluting the extra-vesicular bath with deionized water.

## Data processing

Images were processed using ImageJ—a public-domain software obtained from http://rsbweb.nih.gov/ij/.

## Acknowledgements

We thank A Matysik for technical help. We thank C Bain, O Farago, and N Grønbech-Jensen for illuminating discussions and suggestions. We thank the Expert Reviewers, Editor, and an anonymous critic for their comments, references, and insights, which have helped strengthen the manuscript. This work is supported by a grant from Biomolecular Materials Program, Division of Materials Science and Engineering, Basic Energy Sciences, U. S. Department of Energy under Award # DE-FG02-04ER46173 (ANP). KO and RSK acknowledge support from the Singapore Ministry of Education AcRF Tier II Grant number MOE2009-T2-2-019. KO, ANP, and BL acknowledge additional support from Nanyang Technological University through the Centre for Biomimetic Sensor Science.

## Additional information

### Funding

| Funder | Grant reference number | Author |
| --- | --- | --- |
| U.S. Department of Energy | DE-FG02-04ER46173 | Atul N Parikh |
| Ministry of Education - Singapore | MOE2009-T2-2-019 | Kamila Oglęcka, Rachel S Kraut |
| Nanyang Technological University | Center for Biomimetic Sensor Science | Kamila Oglęcka, Bo Liedberg, Atul N Parikh |

The funders had no role in study design, data collection and interpretation, or the decision to submit the work for publication.

### Author contributions

KO, Conception and design, Acquisition of data, Analysis and interpretation of data, Drafting or revising the article; PR, BL, Analysis and interpretation of data, Drafting or revising the article; RSK, ANP, Conception and design, Analysis and interpretation of data, Drafting or revising the article

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
