## [Decision Letter]

Thank you for sending your work entitled “Oscillatory phase separation in giant lipid vesicles induced by transmembrane osmotic differentials” for consideration at *eLife.* Your article has been favorably evaluated by Randy Schekman (Senior editor) and 3 expert reviewers.

The Senior editor and the reviewers discussed their comments before we reached this decision, and the Senior editor has assembled the following comments to help you prepare a revised submission.

In this manuscript, Ogleca et al. present an extensive study of the behavior of giant liposomes, containing three model lipid species, cholesterol, sphingomyelin and POPC, when subjected to an osmotic stress. The GUVs show periodic oscillations, which includes a swelling phase, formation of Ld and Lo domains, and transient pore formation. With time, the characteristic time period of the oscillations increases until the GUV reaches a quasi-equilibrated state, having gradually relaxed the osmotic stress through transient pore formation. The quality of the study is high: the authors show many controls and the movies and figures nicely support their conclusions. Furthermore, the text is well written and the authors are very careful at citing and discussing all previous studies on the same topics, making the overall study very solid and fair.

The importance of this work is two-fold: i) GUVs are widely used experimental tools. Other than the dedicated membrane labs who focus on these systems, I suspect most users are not well aware of the degree of complexity they can exhibit and the ease with which it can be reached (even accidentally). In this context, I think this work can be highly influential to a broad community of scientists interested in lipid and cell membranes. ii) In a more philosophical sense, this is a beautiful example of a simple chemical system in which multiple order parameters become coupled to produce highly complex (and unexpected to the non-expert) behavior. It underscores the intrinsic coupling between membrane phase and mechanical tension.

The reviewers agree this work should be published in *eLife*, and are happy with the experiments you have conducted, however there are a number of concerns with the text that need addressing in the form of revisions which are enumerated below:

1) The authors did not offer compelling reasons to believe this work has relevance to biological processes. Large domains are seldom observed in cells and current models for domain formation involve complex and transient interactions between not only lipids but also proteins and on a much shorter time scale and also smaller length scale. To my knowledge, the only exception is the formation of large domains in the yeast vacuole (J Cell Biol. 2013 Jul 8;202(1):35-44), which the authors did not discuss. Instead, the authors mention two processes that they say might be biologically relevant:

a) 'we found that domain dynamics become co-ordinated with the previously well-known process of expulsion of internal “organelle” vesicle, reminiscent of vesicle traffic.' I don't see how this process is reminiscent of a biological process. Cells carefully avoid leakage, notably during vesicle trafficking.

b) 'In this same vein, it seems plausible that domain-forming prebiotic amphiphiles might have given a thermodynamic advantage to early, protein-free protocells to survive drastic osmotic environmental osmotic shifts.' Here the authors quote seminal studies by Szostack about plausible lipid mixtures that may have led to primitive cell compartments. However, it should be noted that these lipid mixtures are very far from those studied here. They mostly consist in simple fatty acids, not in diacyl lipids and, as such, are characterized by much higher permeability. In a more recent paper, the Szostack lab explored the advantage of having a few mol % of diacyl lipids, but even under these conditions, the plausible primitive membrane remains very far from what is studied here.

2) Another major problem for the authors is to establish what new knowledge has been gained. No individual component of the behavior they report is previously unobserved, it is important for the authors to clearly state, and convince readers, of what is learned by their observations. In this regard, some of the language they use is perhaps too abstract. For example, the authors write in the Abstract that their observations illustrate a “quasi-homeostatic self-regulatory behavior allowing synthetic vesicles produced from simple components, namely, water, osmolytes, and lipids to sense and regulate their micro-environment in a negative feedback loop”. I'm not sure I see why this is “self-regulatory” or a “negative feedback loop”. It is oscillatory and certainly illustrates coupling between mechanical tension and phase, but I think the authors may stretch things a bit too far and may lose some readers in doing so.

3) “Oscillations in osmotic pressure and membrane tension.” Interspersed among clear explanations are several odd statements such as this one. Membrane tension is certainly oscillating, but osmotic pressure is not. If one were to make a graph of pressure versus time it would be monotonically decreasing as the vesicle equilibrates. As the authors note, this equilibration is more readily achieved by water influx, diluting the interior contents, so tension increases, until high tension drives membrane pore formation. Then, contents leak out, tension drops, and the pressure difference between interior and exterior further drops. Then the pores seal, but since the pressure is nonzero, equilibration continues again via water influx, and the cycle continues; though more slowly, since the driving pressure differential is smaller than it was before. (The authors' data on the slowing is very nice.) All this couples to phase separation, since high tension leads to phase separation, and low tension to homogeneity. Rather than the author's schematic at the end of the paper, a simple schematic of pressure monotonically decreasing and tension oscillating above and below a critical value corresponding to phase separation, would be useful for the reader.

Similarly, the authors refer in a few places to a “negative feedback loop.” This is a bit overstated. As noted above, the system is not oscillating about some equilibrium, rather it is monotonically approaching the equilibrium of zero osmotic pressure, and tension is oscillating as it is doing this. There is, in some sense, a “set point” of tension, the lysis tension, but at equilibrium this also is zero. Moreover, this feedback is related directly only to the tension, and not the phase separation, and so it again seems a bit much to call the overall phenomenon one of self-regulation by negative feedback.

Really what one would like with this is a quantitative model that reproduces the critical lysis tension (see above) or the time between oscillations or both. The paper doesn't provide this. It does contain a nice physical discussion in the supplemental text, but this succeeds only in roughly explaining the pore lifetime. I think it's fine to publish the paper without a real model, but the paper should be shorter than it is by at least 50% - it shouldn't take so many words to explain qualitatively the basic oscillation mechanism outlined above. The paper is very repetitive: the basic explanation for the observed phenomenon is given several times.

In the impact statement and the text it is noted that this phenomenon “of plausible biological significance.” There isn't a strong case made for this. I think this is fine: the paper expands our understanding of what lipid vesicles are capable of, which is important. I think these words should be removed from the impact statement.

[Editors' note: further revisions were requested prior to acceptance, as described below.]

Thank you for resubmitting your work entitled “Oscillatory phase separation in giant lipid vesicles induced by transmembrane osmotic differentials” for further consideration at *eLife.* Your revised article has been favorably evaluated by Randy Schekman (Senior editor) and the original reviewers. The manuscript has been improved but there is one remaining issue that needs to be addressed before acceptance, as outlined below:

Oglecka et al. have done a good job of addressing the concerns that one reviewer had with the original manuscript, with one exception, the area that refers to the original incorrect claim of “oscillations in osmotic pressure and membrane tension.” The authors have included a new schematic illustration “showing cyclical oscillations in tension and stair-case drop in pressure.” The tension schematic is correct; however, the step-wise drops in pressure are not. As stated originally, “If one were to make a graph of pressure versus time *it would be monotonically decreasing* as the vesicle equilibrates. This equilibration is more readily achieved by water influx, diluting the interior contents, so tension increases, until high tension drives membrane pore formation. Then, contents leak out, tension drops, and the pressure difference between interior and exterior *further drops*. Then the pores seal, but since the pressure is nonzero, equilibration continues, again via water influx, and the cycle continues...” The pressure is not stationary at any time. Of course, during the (very brief) leakage period, the rate of pressure drop may be greater than it is during the times between ruptures, but (1) the authors don't make any quantitative argument that this is the case, and (2) even if it were, the qualitative schematic that shows flat pressure lines is quite misleading. Put more simply: the tension is increasing in each cycle ’because’ water flows in, lowering the osmotic pressure. A positive slope for tension occurs together with a negative slope for pressure, and the schematic should show this.

---

## [Author Response]

*1) The authors did not offer compelling reasons to believe this work has relevance to biological processes. Large domains are seldom observed in cells and current models for domain formation involve complex and transient interactions between not only lipids but also proteins and on a much shorter time scale and also smaller length scale*. *To my knowledge, the only exception is the formation of large domains in the yeast vacuole (J Cell Biol. 2013 Jul 8;202(1):35-44), which the authors did not discuss. Instead, the authors mention two processes that they say might be biologically relevant:*

*a) 'we found that domain dynamics become co-ordinated with the previously well-known process of expulsion of internal “organelle” vesicle, reminiscent of vesicle traffic.' I don't see how this process is reminiscent of a biological process. Cells carefully avoid leakage, notably during vesicle trafficking*.

*b) 'In this same vein, it seems plausible that domain-forming prebiotic amphiphiles might have given a thermodynamic advantage to early, protein-free protocells to survive drastic osmotic environmental osmotic shifts.' Here the authors quote seminal studies by Szostack about plausible lipid mixtures that may have led to primitive cell compartments. However, it should be noted that these lipid mixtures are very far from those studied here. They mostly consist in simple fatty acids, not in diacyl lipids and, as such, are characterized by much higher permeability. In a more recent paper, the Szostack lab explored the advantage of having a few mol % of diacyl lipids, but even under these conditions, the plausible primitive membrane remains very far from what is studied here*.

We agree that it is important to be cautious in extrapolating characteristics and behaviors of synthetic vesicles to cells. We do not imply that the specific properties of osmotically generated oscillations in domain morphologies (i.e., sizes, shapes, and temporal frequencies) of giant vesicles would be reproduced in living cells subject to osmotic stresses. A variety of cellular characteristics – including changes in cytoskeleton interactions; size of plasma membrane reservoir activities of mechanosensitive channels and surface area regulation of membranes to phase separate in cells, can all be expected to regulate the osmotic stress and modulate the capacity (Raucher and Sheetz 1999, [55], Morris and Homann 2001). a) Precise sizes and time scales of domain reorganization notwithstanding, the remarkable coordination of essential biophysical processes: (1) osmotically generated membrane tension; (2) osmotic pressure dependent changes in phase separation; (3) transient membrane poration; and (4) solute efflux – which control vesicle osmoregulation should also play a role in guiding the response of a living cell to an osmotic perturbation.

We thank the reviewers for bringing to our attention the case of yeast vacuole (50), which exhibits phase separation of both lipids and proteins in what appears to be a sterol-dependent phase separation into co-existing liquid phases when subject to stresses (i.e., nutrient deprivation, pH changes, and changes in the growth medium) such as seen in synthetic vesicles. We have now incorporated this case (and the corresponding reference) in our narrative discussing potential biological relevance in the revised version of our submission.

Perhaps, one of the most compelling cases for the biological significance of our observations is that many biology labs use phospholipid vesicles and synthetic membranes as tools to investigate membrane-mediated effects in biological processes including cell adhesion, cell-cell interactions, signaling, organelle functions, and mechanisms of viral and bacterial interactions with host cell membranes, to name a few. Appreciating the complexity, heterogeneity, and sensitivity of membrane surfaces to osmotic gradients should prove valuable in interpreting these data and in designing model membrane systems specifically suited to the case at hand. This point is now explicitly made in the revised version of the paper.

We recognize that the parallel we draw between osmotically mediated expulsion of internal vesicles with vesicle trafficking is weak especially since the transient poration in our case is indeed accompanied by solute leakage, which is absent in vesicle trafficking. We have now removed this analogy.

We also recognize that there are significant differences in the properties of vesicles self-assembled from fatty acid amphiphiles, thought to make up pre-biotic membranes and those formed from phospholipids. However, the coupling of osmotic activity with membrane’s internal compositional degrees of freedom is of purely physical and mechanical origins and may be applicable to diverse chemical systems albeit with differences in time and size scales of oscillations. In other words, it appears likely that the heterogeneous pool of fatty acid amphiphiles (including those differing in length of their hydrocarbon tails, degrees of chain saturation, and transition temperature) might also phase-separate and respond to osmotically generated tension (and pressure) by sorting into domains. We are currently pursuing experiments with mixtures of fatty acids to determine if the observations reported here for phospholipid vesicles are also at works in that system.

*2) Another major problem for the authors is to establish what new knowledge has been gained. No individual component of the behavior they report is previously unobserved, it is important for the authors to clearly state, and convince readers, of what is learned by their observations. In this regard, some of the language they use is perhaps too abstract. For example, the authors write in the Abstract that their observations illustrate a “quasi-homeostatic self-regulatory behavior allowing synthetic vesicles produced from simple components, namely, water, osmolytes, and lipids to sense and regulate their micro-environment in a negative feedback loop”. I'm not sure I see why this is “self-regulatory” or a “negative feedback loop”. It is oscillatory and certainly illustrates coupling between mechanical tension and phase, but I think the authors may stretch things a bit too far and may lose some readers in doing so*.

We refer to the vesicle response to an osmotic perturbation in terms of negative feedback loop since the initially isotonic vesicle, when subject to an osmotic differential, undergoes the process of relaxation to minimize the source perturbation. We refer to this process as homeostatic because in so doing, through solute and solvent equilibration, the vesicle changes the bath. We do not mean to suggest that the domain oscillations, coupling mechanical tension and phase, by themselves reflect any sort of feedback loop. We have now rephrased this discussion in the revised version of the manuscript.

Taken individually, it is true that the elemental biophysical processes (i.e., membrane permeability, phase separation, poration, and tension-induced and pressure-dependent changes in membrane organization), which guide vesicle response to osmotic stress are all well-appreciated in extant literature. The co-ordination between these processes, however, producing the emergent behavior in vesicles subject to osmotic difference, appears to us to be quite striking and previously unappreciated. To explicitly state what new knowledge is gained and perspectives earned, we now provide a detailed statement on value and broader implications of our observations:

“Although individual biophysical processes leading to the oscillatory domain dynamics during this osmotic relaxation process are well-appreciated, the observations reported here bring to focus several features of vesicle behaviour, which are best poorly appreciated. First, the seemingly autonomous vesicle response – in which an external osmotic perturbation is managed by a co-ordinated and cyclical sequence of physical mechanisms allowing vesicles to sense (by domain formation) and regulate (by solute efflux) their local environment– suggests a primitive form of a quasi-homeostatic regulation in a synthetic material system (25), i.e., a simple microemulsion produced from simple components, namely, lipids, water, and osmolytes. Second, these observations illustrate how out-of-plane osmotic activity of water becomes coupled with membrane’s in-plane compositional degrees of freedom producing an exquisite and complex response. It underscores the intrinsic coupling between membrane phase and mechanical tension. Third, by highlighting the complexity of lipid vesicles, these results offer important caveats in implementing giant vesicles as experimental models in scenarios where osmotic imbalances can dominate vesicle response.*”*

*3) “Oscillations in osmotic pressure and membrane tension.” Interspersed among clear explanations are several odd statements such as this one. Membrane tension is certainly oscillating, but osmotic pressure is not. If one were to make a graph of pressure versus time it would be monotonically decreasing as the vesicle equilibrates. As the authors note, this equilibration is more readily achieved by water influx, diluting the interior contents, so tension increases, until high tension drives membrane pore formation. Then, contents leak out, tension drops, and the pressure difference between interior and exterior further drops. Then the pores seal, but since the pressure is nonzero, equilibration continue again via water influx, and the cycle continues; though more slowly, since the driving pressure differential is smaller than it was before. (The authors' data on the slowing is very nice.) All this couples to phase separation, since high tension leads to phase separation, and low tension to homogeneity. Rather than the author's schematic at the end of the paper, a simple schematic of pressure monotonically decreasing and tension oscillating above and below a critical value corresponding to phase separation, would be useful for the reader*.

Agreed. We now include a schematic showing cyclical oscillations in tension and stair-case drop in pressure, which occurs with the cycle number as vesicles undergo osmotic equilibration. Because of the value and appeal to broader readership, we also suggest that the original graphical schematic be also retained in the final version of the paper.

*Similarly, the authors refer in a few places to a “negative feedback loop.” This is a bit overstated. As noted above, the system is not oscillating about some equilibrium, rather it is monotonically approaching the equilibrium of zero osmotic pressure, and tension is oscillating as it is doing this. There is, in some sense, a “set point” of tension, the lysis tension, but at equilibrium this also is zero. Moreover, this feedback is related directly only to the tension, and not the phase separation, and so it again seems a bit much to call the overall phenomenon one of self-regulation by negative feedback*.

We concur. As in response to 2) we have rephrased this discussion without implying self-regulation by negative feedback loop as mechanism for the observed coupling of tension and phase in the membranes. We have also included a new schematic in Figure 6 qualitatively highlighting changes in membrane tension, which are oscillatory and in osmotic pressure, which are of the stair-case type.

*Really what one would like with this is a quantitative model that reproduces the critical lysis tension (see above) or the time between oscillations or both. The paper doesn't provide this. It does contain a nice physical discussion in the supplemental text, but this succeeds only in roughly explaining the pore lifetime. I think it's fine to publish the paper without a real model, but the paper should be shorter than it is by at least 50% - it shouldn't take so many words to explain qualitatively the basic oscillation mechanism outlined above. The paper is very repetitive: the basic explanation for the observed phenomenon is given several times*.

We have edited the manuscript further with a goal of minimizing unnecessary repetitions.

*In the impact statement and the text it is noted that this phenomenon “of plausible biological significance.” There isn't a strong case made for this. I think this is fine: the paper expands our understanding of what lipid vesicles are capable of, which is important. I think these words should be removed from the impact statement*.

We have now removed the emphasis on plausible biological significance in the impact statement and more explicitly clarified the coupling of order parameters, which occurs during osmotic equilibration in giant vesicles.

[Editors' note: further revisions were requested prior to acceptance, as described below.]

*Oglecka et al. have done a good job of addressing the concerns that one reviewer had with the original manuscript, with one exception, the area that refers to the original incorrect claim of “oscillations in osmotic pressure and membrane tension.” The authors have included a new schematic illustration “showing cyclical oscillations in tension and stair-case drop in pressure.” The tension schematic is correct; however, the step-wise drops in pressure are not. As stated originally, “If one were to make a graph of pressure versus time* it would be monotonically decreasing *as the vesicle equilibrates. This equilibration is more readily achieved by water influx, diluting the interior contents, so tension increases, until high tension drives membrane pore formation. Then, contents leak out, tension drops, and the pressure difference between interior and exterior* further drops*. Then the pores seal, but since the pressure is nonzero, equilibration continues, again via water influx, and the cycle continues...” The pressure is not stationary at any time. Of course, during the (very brief) leakage period, the rate of pressure drop may be greater than it is during the times between ruptures, but (1) the authors don't make any quantitative argument that this is the case, and (2) even if it were, the qualitative schematic that shows flat pressure lines is quite misleading. Put more simply: the tension is increasing in each cycle ’because’ water flows in, lowering the osmotic pressure. A positive slope for tension occurs together with a negative slope for pressure, and the schematic should show this*.

This was a clearly a gross oversight on our part. Thank you for bringing it to our attention. The osmotic influx of water prior to pore formation, which renders the membrane tense, also acts to reduce the osmotic pressure by diluting the osmolyte concentration. Solute efflux during the short-lived poration event provides a second, separate mechanism by which pressure drops. Thus during the entire osmotic equilibration process pressure drops monotonically. Although we have not quantitatively compared the pressure drop due to dilution and that occurs during solute efflux, we surmise, based on a phenomenological analyses following Idiart and Levin (26, 30), that pressure drop is more drastic during solute efflux.

We have now rectified the pressure schematic in Figure 7 and noted in the figure legend that the relative rates are conjectures, and further quantitation is needed.